# Care use and its intensity in children with complex problems are related to varying child and family factors: A follow-up study

**Noortje M. Pannebakker**[1,2]*, **Paul L. Kocken**[1,2], **Paula van Dommelen**[1], **Krista van Mourik**[2], **Ria Reis**[2], **Sijmen A. Reijneveld**[3], **Mattijs E. Numans**[4]

**1** Department of Child Health, TNO, Leiden, Netherlands, **2** Department of Public Health and Primary Care, Leiden University Medical Center, Leiden, Netherlands, **3** Department of Health Sciences, University of Groningen, University Medical Center Groningen, Groningen, Netherlands, **4** Department of Public Health and Primary Care and LUMC Campus The Hague, The Hague, Netherlands

* noortje.pannebakker@tno.nl

**Data Availability Statement:** We are not able to make all data underlying the findings described in our manuscript fully available without restriction,

## Abstract

### Background

There is little evidence on the child and family factors that affect the intensity of care use by children with complex problems. We therefore wished to identify changes in these factors associated with changes in care service use and its intensity, for care use in general and psychosocial care in particular.

### Methods

Parents of 272 children with problems in several life domains completed questionnaires at baseline (response 69.1%) and after 12 months. Negative binominal Hurdle analyses enabled us to distinguish between using care services (yes/ no) and its intensity, i.e. number of contacts when using care.

### Results

Change in care use was more likely if the burden of adverse life events (ALE) decreased (odds ratio, OR = 0.94, 95% confidence interval, CI = 0.90–0.99) and if parenting concerns increased (OR = 1.29, CI = 1.11–1.51). Psychosocial care use became more likely for school-age children (vs. pre-school) (OR = 1.99, CI = 1.09–3.63) if ALE decreased (OR = 0.93, CI = 0.89–0.97) and if parenting concerns increased (OR = 1.26, CI = 1.10–1.45). Intensity of use (>0 contacts) of any care decreased when ALE decreased (relative risk, RR = 0.95, CI = 0.92–0.98) and when psychosocial problems became less severe (RR = 0.38, CI = 0.20–0.73). Intensity of psychosocial care also decreased when severe psychosocial problems became less severe (RR = 0.39, CI = 0.18–0.84).

### Conclusions

Changes in care-service use (vs. no use) and its intensity (>0 contacts) are explained by background characteristics and changes in a child's problems. Care use is related to factors

because the informed consents given by our participants do not give us the permission to do so. We are able to provide a detailed metadata description of our dataset and we have made all our methods and analysis scripts available within the DANS repository at DOI: 10.17026/dans-xnr-j7yj. Researchers interested in our dataset can contact us via our data access committee. The data access committee is responsible for reviewing applications and finding the appropriate solution for data access in agreement with the given informed consents. We are happy to supply the contact information of our data access committee: Name: Front Office TNO Child health Email: zelfstandig-tno-secretariaatCHLS@tno.nl Phone number: 0031+(0) 88 866 90 00 Address: Schipholweg 77 2316 ZL Leiden P.O. 3005, 2301 DA Leiden

**Funding:** This study was funded by ZonMw, The Netherlands Organization for Health Research and Development (15901.0005). This funding body played no role in the design, conducting, analysis or write-up of the study.

**Competing interests:** The authors have declared that no competing interests exist.

other than changes in its intensity, indicating that care use and its intensity have different drivers. ALE in particular contribute to intensity of any care use.

## Introduction

Little research has been conducted on factors affecting the intensity of care use by children with complex problems (CP*). The need of these* children for health *and social* services, especially psychosocial care, is typically greater than would be expected *based on* their chronic physical, developmental, behavioral or emotional conditions; *this is the case because their problems interact and enhance vulnerabilities* [1–4]. *These children are also referred to as members of troubled families or hotspotters* [5–7]. *Children with complex problems form the top 5% of children with the most challenging problems, amounting in the Netherlands to 170,000 children. Western countries struggle to organize effective and efficient care pathways for these children* [8]. *As a result, a major part of the budgets of psychosocial services is spent on children with CP* [9,10].

*The determinants of care use as such have been studied in depth, revealing several factors that impact access* [11–16]. *Less attention was paid to understanding the intensity of care use*, i.e. the number of contacts with care providers. The scarce literature shows that higher intensity of care-service use by children with CP is related to two main groups of factors: child factors (age and impact of psychosocial problems); and parental factors (educational level, healthcare use, social support, and parental psychosocial problems) [17–20]. *Research shows that the determinants affecting care use (yes/no) and the intensity with which it is used differ when studied simultaneously* [17–19]. *This suggests that intensity of care use may be a unique component of the help-seeking behavior of families with a child with complex problems. A better understanding of the intensity of care use will help us to organize more efficient care paths for these children.*

Research on determinants of care use is often guided by Andersen and Newman's behavioral-health model [21]. *This model was developed to explain the use of care by an individual or population and has shown its value as comprehensive model for this purpose in health care research during the past decades [22,23].* The model describes care use on the basis of three factors: 1. predisposing factors, i.e., a child's characteristics or abilities to use a specific service (such as age); 2. enabling factors, i.e., means whereby a family accesses care (such as social support); and 3. healthcare needs (such as a child's psychosocial problems). This broad framework is a good fit with the wide-ranging problems experienced by children with CP. Our study is the first to apply this framework to the intensity of care use by these children.

We previously reported that overall care use *was associated with social support and psychosocial problems and that the use of psychosocial care was associated with a child's age and parenting concerns, based on* a cross-sectional study in families with severe complex problems [16]. In the current study with a follow-up design, *we additionally examined changes in intensity of care use in children with and at risk on developing CP, ensuring a wide range of intensity of care use. Accordingly,* the aim of this study is to identify the changes in the predisposing, enabling and need factors that are associated 1. with a higher likelihood of changes in use of care services and 2. with changes in the intensity of use. The care services use studied comprised *a broad spectrum of general care services including health and psychosocial care,* and also the subset psychosocial care, *including child mental healthcare and child and family services. We selected several predisposing characteristics of the child (such as age and gender, and also including predetermined factors such as parental education level and adverse life events), as well*

*as enabling (social support and parental care use) and need factors (chronical condition, psychosocial problems, satisfaction with the parent-child relationship, and parenting concerns). This selection was based on the literature regarding determinants impacting the intensity of use of psychosocial care by children, as well as on our former study* [4–6; 11–16].

# Method

## Sample and procedure

For this longitudinal study, we followed a cohort of children with CP and their parents, living in an urban setting in the Netherlands. The study was conducted according to the Helsinki regulation. The Medical Ethics Committee at Leiden University decided that approval was not required under Dutch Law (C12.041).

*We aimed to include* parents of children with CP or at risk of developing them with a wide range in intensity of care use living in the community. We recruited these *parents in the general population, using inclusion criteria that concurred with the framework for identifying families with CP [24,25]. We included parents when they met the following inclusion criteria*: 1. they had a child between 18 months and 12 years and 2. they experienced at least one of the following conditions: A. the child's elevated total score on the parent-reported Strengths and Difficulties Questionnaire (SDQ) [26] or Brief Infant Toddler Social Emotional Assessment (BITSEA) [27]; B. persistent parenting concerns as judged by the preventive health care worker and/or parents; C. one or more major life event(s) during the past year as assessed using the *standard screening questionnaire of the well child clinic [28]* and D. care utilization of the child or parent in the past six months. Almost all respondents had three or more of these conditions (97%).

We identified *the respondents* during well-child visits, which are provided in the Netherlands by the preventive youth healthcare services. Attendance rates at these visits are high: 95% of all children [29]. To ensure the inclusion of children who used care with a high intensity, we additionally included children enrolled in specialist child and family services *i.e. services that are only accessible after referral from primary care. Together this study group is expected to represent the whole group of children with CP or at risk of developing them.*

*We used the following inclusion procedure. First, a nurse, doctor or social worker identified parents based on our inclusion criteria, which were embedded in their routine intake questionnaire. Professional care givers then provided oral and written information about the study to the identified families and asked permission from parents to be called by a research assistant. Thereafter, the research assistant asked for informed consent regarding participation in this study.*

Data were collected *by trained research assistants* at two time points, the first in 2013 (T1) and the second 12 months later (T2). Data were collected in a digital questionnaire, although parents could also opt to be interviewed by telephone in the language of their preference. *Parents were reminded three times to fill in this questionnaire and received a gift certificate of 20 euros after doing so. Parents were informed that they could withdraw at any moment.*

A total of 512 parents were approached, 354 of whom participated at T1 (response = 69.1%). Of these, 272 participated in the follow-up at 1 year (T1-T2 response = 76.8%), 239 from the well-child clinic group (T1 = 309), and 33 from the group of children using care in a high intensity (T1 = 45). Parents who dropped out at T2 had significantly more sons; more of them were of non-western origin and, on the basis of their home neighborhoods, and more of them had a lower socio-economic position [30].

## Measures

*We used validated questionnaires if available and assessed their reliability in the sample under study.* The children's service use *and intensity of this use* in the past six months were measured with the Q*uestionnaire Intensive Care for Youth, a questionnaire measuring use of a pre-set list of types of Dutch services [31, 32]. This list has been adapted to the setting of care for youth from the valid and reliable Questionnaire* for Costs Associated with Psychiatric Illnesses and Care Use *(TiC-P) [33,34]. As allowed for by this standard questionnaire, we have added and omitted specific items of care services depending on their relevance for our target population.* Moreover, *respondents had the opportunity to add services we had not listed. Services are defined as any care provider or group of care providers.* Dichotomized use at baseline and at follow-up led to four categories expressing change in use, i.e. "never used care", "stopped using care", "started using care" and "continued using care". Intensity of care service use was measured as the number of contacts, defined as planned or unplanned contacts with a professional caregiver by telephone, email, or appointment or home visit; this did not include contacts to make an appointment. We made a distinction between 1. use of any services, which included the use of care delivered in the psychosocial or medical domain; and 2. use of psychosocial services, which included a subset of any care delivered by mental healthcare services, social care services, school care services or family services.

On the basis of Andersen and Newman's behavioral-health model of access to care, we measured potential determinants of care use, i.e. predisposing, enabling and need factors [21]. We used six predisposing factors: child's age; parents' educational level; household composition; child's ethnicity; parental mental health, and impact of any adverse life events the family had experienced (ALE). Parental mental health status was measured using *the validated* 12-item version of the General Health Questionnaire (Cronbach's α = .86) [35]. To measure the burden of adverse life events in the previous 12 months, we used the life-events scale of the Brief Instrument Psychological and Pedagogical Problem Inventory (Cronbach's α = .72) [36].

We measured three enabling factors: partner's provision of social support, family provision of social support, and care use by a parent. To measure social support, we used two subscales of the *validated Dutch* Family Functioning Questionnaire [37]: "relationship with partner"(-Cronbach's α = .88), and "social functioning of the family" (Cronbach's α = .91). Parental care use was measured using the TiC-P, similar to the way the child's care use was measured (see above) [33].

We included four need factors in this study, i.e. a child's chronic condition; a child's emotional and behavioral problems; parenting concerns and a parent's assessment of the quality of their relationship with their child. Questions measuring a child's chronic health were the following. "Does your child have one or more chronic health condition—such as asthma, diabetes, ADHD or autism—for which treatment is or was needed? What is the impact of this condition on your child's daily life?" *[38].* We measured child behavioral and emotional problems using the BITSEA (for children aged between 18 months and 3 years) and the SDQ (for children aged between 3 and 12 years). We constructed the variable as a dichotomy, to be able combine the scores on the different instruments for the whole group. The Dutch versions of both were found to be reliable [26, 39–42]. Cronbach's α's *as measured in this study* of the SDQ subscales range from .39 to .74 and Cronbach's α's of the two BITSAE subscales on our data were .67 and .80. To measure parenting concerns we used the following question: "In the last 12 months, have you had concerns about your parenting?"[43] Finally, parents' assessment of their relationship with their child was measured using the subscale of the *validated Dutch* Parenting Load Questionnaire (Cronbach's α = .83) [44]. *Answers were given in a five point Likert-scale. We dichotomized scale sum scores using their medians as cut off.*

## Analyses

First, we described the background characteristics; the scores of the predisposing, enabling and need factors; the use and intensity of any care; and the use and intensity of psychosocial care, all at baseline (T1). Next, on the basis of patterns of use and intensity, we described the changes between baseline and follow-up of predisposing, enabling and need factors for use of any care and use of psychosocial care. Next, we used negative binomial Hurdle modeling to assess the associations between the changes in factors and the changes in care use and its intensity. The score of the dependent variable at T1 was entered as covariate in the Hurdle analyses to be able to address 'change' in the outcomes.

Our use of Hurdle modeling was intended to overcome the statistical challenges inherent to data on care use, which typically follow a distribution with many zeroes (no use of care) [45, 46]. Hurdle models have the advantage of estimating two separate parameters in one model to accommodate many zero counts: one dichotomous outcome regarding using care services or not (>0 contacts versus no contacts), and one continuous outcome regarding the number of contacts within the group using care services (>0 contacts). First we assessed the univariate associations of the changes in the independent variables with any care and psychosocial care. On the basis of backward elimination in Hurdle models of all independent variables that were univariately significantly related at the p≤0.1 level, we then assessed which predisposing, enabling and need factors, and changes in these, were associated with changes in use and its intensity. The criterion for removing a factor out from the final models was set at p≤ 0.05. Analyses were performed using SPSS 22.0 [47], and the Hurdle analyses were performed in R, version 3.3.2 [48].

## Results

### Response and respondents' background characteristics

The sample included more boys than girls, more school-aged children than pre-schoolers, more children of two-parent families than one-parent families, and more children of Dutch ethnicity than of non-Dutch ethnicity (see Table 1). About half of the parents with a high educational level experienced mental health problems and/or *experienced* burden of adverse life events in the previous year.

**Care service use and its intensity, and scores on predisposing, enabling and need factors.** At baseline, three-quarters of the children in our sample were using some sort of care, and 45% were using psychosocial care (Table 1). For any child using care services, the average intensity of care use was 21 contacts. The intensity of service use was higher if a child was using psychosocial care, with an average of 25 contacts in the previous six months. Predisposing and enabling factors followed a different pattern for 'any' and 'psychosocial care' use (yes/no) and their intensity (>0 contacts). However, need factors showed the same pattern both for any care use and for psychosocial care use: children whose parents reported a higher score on a need factor used care more often and with a higher intensity than those who reported a lower score.

**Change in a child's predisposing, enabling and need factors for care service use and its intensity.** Table 2 shows changes in predisposing, enabling and need factors and in care use (yes/no) and its intensity (>0 contacts) over time during the use of care services. Regardless of their difference score on the independent variable, most children were in the "continued care use" category for any care services, and in the "never used care" category for psychosocial care services. More children with an increase in the level of need factors tended to be in the "started care" category than those whose needs were decreased or remained unchanged.

**Table 1. Respondents' baseline characteristics, and care use and intensity of use of any care and of psychosocial care.**

| | Total | Any care services | | Psychosocial care services | |
|---|---|---|---|---|---|
| | N## | any use<br>n (%)ª | intensity when using mean (SD)ᵇ | any use<br>n (%)ª | intensity when using mean (SD)ᵇ |
| Total | 272 | 203 (75) | 21 (35) | 121 (45) | 25 (39) |
| Predisposing factors | | | | | |
| Child's gender | | | | | |
| Boy | 152 | 117 (77) | 24 (41) | 67 (55) | 29 (46) |
| Girl | 120 | 86 (72) | 17 (25) | 54 (45) | 20 (29) |
| Child's age | | | | | |
| Pre-school | 107 | 67 (63) | 25 (41) | 38 (36) | 28 (44) |
| School-aged | 165 | 111 (67) | 15 (23) | 87 (53) | 15 (19) |
| Parental educational level | | | | | |
| High | 132 | 93 (71) | 24 (39) | 53 (40) | 18 (26) |
| Low/ medium | 138 | 109 (79) | 16 (26) | 67(49) | 28 (43) |
| Household composition | | | | | |
| 2-parent family | 133 | 109 (82) | 19 (34) | 55 (41) | 27 (39) |
| 1-parent family | 112 | 73 (65) | 22 (37) | 49 (44) | 24 (45) |
| Other | 23 | 19 (83) | 26 (30) | 15 (65) | 20 (18) |
| Ethnicity | | | | | |
| Dutch | 155 | 122 (79) | 22 (38) | 78 (50) | 25 (48) |
| Western | 24 | 16 (67) | 17 (24) | 11 (46) | 17 (25) |
| Non-Western | 91 | 64 (70) | 20 (33) | 31 (34) | 28 (38) |
| Parent had mental health problems | | | | | |
| Yes | 150 | 91 (75) | 23 (42) | 67 (45) | 30 (47) |
| No | 122 | 112 (75) | 18 (25) | 54 (44) | 19 (26) |
| Burden of adverse life events | | | | | |
| High | 133 | 100 (76) | 26 (40) | 64 (48) | 30 (44) |
| Low | 124 | 94 (75) | 16 (31) | 51 (41) | 19 (35) |
| Enabling factors | | | | | |
| Partner's provision of social support | | | | | |
| High | 140 | 105 (75) | 25 (40) | 53 (38) | 22 (36) |
| Low | 130 | 97 (75) | 17 (31) | 67 (52) | 28 (42) |
| Family provision of social support | | | | | |
| High | 139 | 110 (79) | 19 (31) | 70 (50) | 18 (31) |
| Low | 130 | 90 (69) | 24 (40) | 51 (39) | 35 (47) |
| Care use by parent | | | | | |
| Yes | 110 | 88 (80) | 26 (38) | 54 (49) | 30 (40) |
| No | 162 | 115 (71) | 17 (38) | 67 (41) | 21 (39) |
| Need factors | | | | | |
| Burden due to chronic condition | | | | | |
| Yes | 53 | 44 (83) | 41 (53) | 32 (60) | 44 (55) |
| No condition/ no burden | 217 | 159 (73) | 25 (26) | 89 (41) | 18 (29) |
| Psychosocial problems | | | | | |
| Yes | 124 | 100 (81) | 26 (42) | 64 (52) | 32 (46) |
| No | 138 | 95 (69) | 14 (21) | 53 (38) | 13 (16) |
| Parenting concerns | | | | | |
| High | 120 | 90 (75) | 30 (44) | 60 (50) | 35 (47) |
| Low | 152 | 113 (74) | 14 (24) | 61 (40) | 15 (28) |

*(Continued)*

**Table 1.** (Continued)

| | Total | Any care services | | Psychosocial care services | |
|---|---|---|---|---|---|
| | N## | any use | intensity when using mean (SD)b | any use | intensity when using mean (SD)b |
| | | n (%)a | | n (%)a | |
| Parental satisfaction with parent-child relationship | | | | | |
| High | 144 | 115 (80) | 33 (19) | 61 (42) | 22 (36) |
| Low | 127 | 88 (69) | 23 (39) | 60 (47) | 28 (43) |

## N is taken at T1 and n varies due to missing data.

a Respondents using care and the within group percentage.

b Mean and standard deviation of care contacts when a respondent was using care.

Regarding change in the intensity of care service use (>0 contacts), we found in the "continuing care" category that children whose need factors had decreased also showed a decrease in the intensity of care use. In line with this finding, children with an increase in need factors also showed an increase in the intensity of care in the "continuing care" category. However, we did not find the same relationship for parenting concerns.

Table 3 shows the final Hurdle models regarding the multivariate associations of change in independent variables with change in use of any and psychosocial care, and their intensity (see S1 Appendix for the results of the univariate regression models for all factors). First we discuss the zero part of the Hurdle models regarding care use (yes/no). The final model regarding use of any care consisted of burden of adverse life events (ALE) (a predisposing factor), and parenting concerns (a need factor). Whereas a decrease in ALE was associated with lower odds of change of care use, an increase in parenting concerns was associated with higher odds. Regarding psychosocial care use, the final model consisted of the same factors as use of any care, but with the addition of child's age. School-aged children had higher odds on change of psychosocial care use than did pre-school children.

Next, we discuss the count part of the Hurdle analyses (>0 contacts). Regarding the intensity of any care use, the final model consisted of burden of adverse life events (a predisposing factor), and psychosocial problems (a need factor). A child's decrease in ALE was associated with decreased intensity of use, and a child's decrease in psychosocial problems was associated with decreased intensity of psychosocial care, in comparison with children with no changes in their level of problems. The final model for psychosocial care services consisted only of psychosocial problems (a need factor), with associations similar to those for any care.

## Discussion

This study shows that changes in the predisposing and need factors of Andersen and Newman's behavioral-health model of access to care were relevant to explaining changes in care use and its intensity by children with CP or at risk of developing it. However, enabling factors were not. We also found that care use was related to factors other than changes in its intensity. Relative to the situation at baseline, when children experienced a diminished burden of life events (ALE) or when more parenting concerns were reported at follow-up, children were less likely to use any care or psychosocial care. School-aged children were also more likely than pre-schoolers to use psychosocial care. The intensity of any care use and of psychosocial care use decreased when the degree of psychosocial problems decreased. The intensity of any care use also decreased when ALE decreased. Moreover, where ALE was associated both with care use and with its intensity, parenting problems uniquely impacted care use and psychosocial problems uniquely impacted its intensity.

**Table 2. Descriptives for the change in predisposing, enabling and need factors and any and psychosocial care use and its intensity by children with or at risk of developing CP.**

| | N# | Care service use (yes/no) | | | | Intensity/number of contacts when using care | | | |
| --- | --- | --- | --- | --- | --- | --- | --- | --- | --- |
| | | never used care | stopped using care | started using care | continued using care | stopped using care | started using care | continued using care | |
| | | | | | | | | T1 | T2 |
| | | n (%)[a] | n (%)[a] | n (%)[a] | n (%)[a] | mean (sd)[b] | mean (sd)[b] | mean (sd)[b] | mean (sd)[b] |
| *Results for any care* | | | | | | | | | |
| Predisposing factors | | | | | | | | | |
| Δ Burden of adverse life events | | | | | | | | | |
| No change | 181 | 21 (12) | 38 (21) | 16 (9) | 106 (59) | 7 (8) | 19 (32) | 25 (44) | 19 (35) |
| Decrease | 57 | 10 (18) | 7 (12) | 10 (18) | 30 (53) | 38 (15) | 9 (9) | 26 (30) | 27 (35) |
| Increase | 9 | 2 (22) | 2 (22) | 0 | 5 (56) | 8 (10) | - | 8 (11) | 16 (13) |
| Need factors | | | | | | | | | |
| Δ Child's psychosocial problems | | | | | | | | | |
| No change | 198 | 29 (15) | 39 (20) | 22 (11) | 108 (55) | 9 (12) | 15 (28) | 22 (34) | 21 (30) |
| Decrease | 39 | 9 (23) | 7 (18) | 3 (8) | 20 (51) | 48 (59) | 4 (5) | 29 (34) | 10 (10) |
| Increase | 21 | 1 (5) | 2 (10) | 3 (15) | 15 (70) | 10 (9) | 15 (6) | 21 (15) | 35 (64) |
| ΔParenting concerns | | | | | | | | | |
| No change | 197 | 26 (13) | 37 (18) | 17 (9) | 117 (59) | 10 (13) (911) | 17 (32) | 26 (41) | 19 (24) |
| Decrease | 17 | 3 (18) | 3 (18) | 2 (12) | 9 (53) | 5 (3) | 10 (11) | 20 (22) | 58 (92) |
| Increase | 57 | 11 (19) | 13 (23) | 10 (18) | 23 (40) | 14 (24) | 8 (8) | 20 (36) | 16 (30) |
| *Results for psychosocial care* | | | | | | | | | |
| Predisposing factors | | | | | | | | | |
| Child's age[d] | | | | | | | | | |
| Pre-school | 107 | 54 (50) | 15 (14) | 21 (20) | 17 (16) | 14 (22) | 8 (13) | 17 (16) | 19 (18) |
| School-aged | 165 | 51 (31) | 27 (16) | 25 (15) | 62 (38) | 12 (11) | 10 (12) | 36 (51) | 22 (30) |
| Δ Burden of adverse life events | | | | | | | | | |
| No change | 181 | 70 (39) | 31 (17) | 31 (17) | 49 (27) | 8 (8) | 10 (14) | 38 (54) | 23 (32) |
| Decrease | 57 | 19 (33) | 4 (7) | 12 (21) | 22 (39) | 36 (10) | 7 (8) | 25 (30) | 19 (22) |
| Increase | 9 | 4 (44) | 2 (22) | 0 | 3 (27) | 7 (9) | - | 14 (12) | 13 (14) |
| Need factors | | | | | | | | | |
| Δ Child's psychosocial problems[c] | | | | | | | | | |
| No change | 198 | 78 (39) | 30 (15) | 33 (17) | 57 (29) | 11 (12) | 12 (14) | 26 (36) | 22 (30) |
| Decrease | 39 | 16 (41) | 6 (15) | 7 (18) | 10 (26) | 8 (8) | 3 (3) | 44 (56) | 13 (11) |
| Increase | 21 | 7 (33) | 3 (15) | 4 (19) | 7 (33) | 30 (41) | 8 (4) | 15 (18) | 28 (24) |
| Δ Parenting concerns[c] | | | | | | | | | |
| No change | 197 | 75 (38) | 28 (14) | 28 (14) | 66 (34) | 13 (13) | 12 (15) | 31 (48) | 20 (22) |
| Decrease | 17 | 6 (35) | 3 (18) | 4 (23) | 4 (23) | 3 (2) | 6 (3) | 27 (29) | 57 (82) |
| Increase | 57 | 24 (42) | 10 (18) | 14 (24) | 9 (16) | 15 (12) | 6 (8) | 37 (46) | 13 (19) 14 (87) |

## n varies due to missing data.

[a] Respondents using care and the within group-percentage.

[b] Mean and standard deviation of care contacts when a respondent who used care.

**Table 3. Final Hurdle models for change in factors associated with change in care use and its intensity by children with CP using care: Multivariate odds ratios for changes in care use and rate ratios for changes in intensity of care use for any care and for psychosocial care services.**

| | Δ Care service use (yes/no) | Δ Intensity/number of contacts when using care |
|---|---|---|
| | adj. OR (95% CI)[ab] | adj. RR (95% CI)[ac] |
| **Final model for any care**[>>] | | |
| Predisposing factors | | |
| Δ Burden of adverse life events[d] | 0.94 (0.90;0.99)* | 0.95 (0.92;0.98)** |
| Need factors | | |
| Δ Child's psychosocial problems[d] | | |
| No change | Ref (1) | Ref (1) |
| Decrease | 0.73 (0.32;1.68) | 0.38 (0.20;0.73)** |
| Increase | 3.27 (0.69;15.48) | 1.17 (0.54;2.56) |
| Δ Parenting concerns[c] | 1.29 (1.11;1.51) *** | 1.13 (0.99;1.29) |
| **Final model for psychosocial care**[>>] | | |
| Predisposing factors | | |
| Child's age[d] | | |
| Pre-school | Ref (1) | Ref (1) |
| School-aged | 1.99 (1.09;3.63)* | 1.32 (0.72;2.43) |
| Δ Burden of adverse life events[d] | 0.93 (0.89;0.97)*** | 0.98 (0.95;1.01) |
| Need factors | | |
| Δ Child's psychosocial problems[e] | | |
| No change | Ref (1) | Ref (1) |
| Decrease | 0.84 (0.36;1.97) | 0.39 (0.18;0.84)* |
| Increase | 1.02 (0.36;2.92) | 1.16 (0.46;2.90) |
| Δ Parenting concerns[c] | 1.26 (1.10;1.45)** | 1.08 (0.95;1.24) |

[a] Backward stepwise regression analyses were conducted with the difference score of the factor, if available, and care use at T1 as covariate. The factors entered were parental educational level, child's age, burden of adverse life events, partner's provision of social support, child's chronic condition, child's psychosocial problems, and parenting concerns. The criterion for removing a factor from the model was set at P-value>0.05

[b] Predictors were removed in the following order: chronic condition, parental educational level, and partner's provision of social support

[c] Only one factor, chronic condition, was removed from the model.

[d] These factors are constructed as difference-of-scale scores between T2-T1.

[e] This factor is constructed as difference of dichotomized scores between T2-T1.

*p<0.05

**p<0.01

***p<0.001.

We found that several changes in predisposing *(i.e. burden of ALE and a child's age)* and need factors *(i.e. parenting concerns and psychosocial problems)* were associated with changes in care use and its intensity, both for overall care use and for the use of psychosocial services, *but that changes in enabling factors were not.* The determinants we found are in line with previous findings *[49–55].* An explanation may be that enabling factors are harder to change than predisposing and need factors *in the relative short time span of our study (one year). For example, it is more difficult for a child social worker to convince parents to make use of mental health care for their own mental problems than to address parenting concerns.* This study shows the value of the Andersen and Newman model for studying the intensity of care use, especially in

distinguishing enabling factors from other factors affecting families with a child at risk of CP or of developing them.

The results of this study added burden of ALE as a factor impacting change in intensity of care use. *Research showed that ALE is an important determinant of care use in general [56, 57]. ALE will especially affect children with CP, interacting strongly with the other problems of these children, thereby leading to more intense problems.* Unexpectedly, we found a slight negative relative risk between ALE and the intensity of any care use. *We noted that the burden of ALE decreased in a relatively large group of children while they were using care. Children with CP may have been motivated to continue treatment even when the burden of ALE decreased, because trauma-based therapies are known to have a positive effect on other emotional conditions [58,59]. Furthermore, when the safety of a child is at risk, as in cases of domestic violence, care professionals will ideally continue treatment to monitor the situation.* Our results indicate that change in ALE is relevant to the whole care process, i.e. not only care use itself, but also to its intensity.

*Although improving social support is at the core of treatment of families with complex problems, in the final models of our study this factor was absent [60]. It can be hypothesized that social support works differently for families with complex problems than for the general population [16, 61–63]. The families' social networks in case of CP are usually large and suitable for dealing with daily challenges of living with a child with complex problem[61–63]. However, regardless of their perceived social support, families will turn to professionals to bring about long-term improvements, surmising that they may not be able to achieve these improvements with their own network. Also, professionals may not yet have managed to bring about changes in the quality of support by the social environment because of the relatively short period of our study (one year).* Although social support is a known determinant impacting a child's care use, more research is needed to understand how to optimize its impact for families of a child with CP.

Finally, we found that changes that changes in intensity with which care is used (>0 contacts) were affected by factors other than changes in care use in itself (yes/no). This supports earlier findings in the scarce research available on intensity of care use [8–10]. For both any care and psychosocial care, our study shows that parents with parenting concerns were more likely to use care, and the intensity of care use increased when there were psychosocial problems. Both need factors are known drivers of help-seeking behavior [64]. Our study showed that parenting concerns impacted care use but not intensity, while a child's psychosocial problems were relevant to intensity rather than to care use.

## Strengths and limitations

A major strength of this study is its comprehensive use of the data, obtained by using the Hurdle model. This model overcomes the difficulties inherent to using a single model to assess factors that impact care use and its intensity, which cannot be assessed by mainstream generalized linear models. We therefore believe that the use of Hurdle models provides added value for researchers interested in care utilization. *Another strength of the study is that the study group of children with or at risk of developing CP were living in the community, including children in treatment with different intensities of care use or not using care at all. In most other research the study groups are limited to children with CP who are using a specific treatment [8,25].*

A limitation of this study concerned some small selective loss to follow-up. A relatively high number of children who were lost to follow-up were boys and had parents of non-western origin. *Another limitation is that we used a self-report questionnaire to establish care use in the previous six months. This may have caused some recall bias, especially for intensity of care use and the determinants burden of ALE and impact of chronical conditions of the child. This may have*

*added measurement error and thus a weakening of reported associations, probably without clear under- or overestimation.*

## Implications for practice

A new finding in this study is the effect that the burden of ALE has on the intensity of care use, a factor that is relevant to the whole care-seeking process, i.e. not only entering care, but also the intensity of its use. This shows the importance of providing interventions that focus on the effects of ALE, on the impact of these effects on intensity, and thus on the costs of care [58, 59]. For this reason, those who assess and treat children with CP should pay close attention to adverse life events and the way children and their families deal with them.

We also found that, while a decline in psychosocial problems was associated with a decrease in intensity of care use, care use in itself was not affected by changes in psychosocial problems. Conceivably, various barriers hinder the process of starting care. In their recent systematic review, in which they provide an overview of the barriers facing children with or at risk of developing CP, Reardon and colleagues show how insufficient knowledge and understanding of psychosocial problems and the help-seeking process on the part of parents is a core component that hinders care use [65]. Policymakers and professional care providers should make efforts to educate parents on recognizing their child's psychosocial problems, and also on the local pathways to help.

## Implications for further research

With regard to care use and its intensity in this group of children, our study shows the enabling factors defined by Andersen and Newman to be less relevant than the predisposing and need factors [8]. To understand the contribution and any possible indirect impact of enabling factors, further research is required. We therefore have two recommendations: 1. a larger respondent group (to accommodate mediation analysis); and 2. extension of the time-lapse in the longitudinal design.

*Regarding the enabling factor social support, more research is needed on how to improve the quality of support provided by the network of families with a child with complex problems. We advise the development and evaluation of a treatment module for parents and key persons in their social network to improve support skills. These new skills can be thought by volunteers who are able to model healthy support.*

In this group of children we also found that the intensity of use of care services is affected by factors different from those influencing the use of care in itself. Understanding the mechanism underlying the intensity of care use can help the *development of more effective and efficient pathways to care for children with or at risk of developing CP*. This will require further research into this mechanism behind care use and its intensity by children with CP.

## Conclusion

With regard to the use of any care, or psychosocial care, and the intensity of this care by children who with or at risk of developing CP, our study shows that changes in predisposing factors (i.e., a child's age and burden of life events) and need factors (i.e., a child's psychosocial problems and parenting concerns) are associated with change in use or intensity of use, and enabling factors are not. The importance of effective treatment of ALE is emphasized by the fact that ALE are a factor that contributes to the intensity of care use. The level of a child's psychosocial problems is also relevant to the intensity of care use (>0 contacts), but not to the use

of care in itself (yes/no). To improve care use by children with these needs, policymakers should address parents' knowledge with regard to identifying psychosocial problems and the help-seeking process. Finally, our findings demonstrate the added value of studying the intensity of care use, especially on the basis of Andersen and Newman's model of care-seeking. Such study will improve our insight into the drivers of the intensity of care use by children with CP.

## Supporting information

**S1 Appendix. Results of the hurdle analyses.**
(DOCX)

**S2 Appendix. Interview protocol for the project oké in Den Haag.**
(DOCX)

**S1 File.**
(PDF)

**S2 File.**
(DOCX)

## Acknowledgments

First and foremost we thank the parents who participated in this study. Thanks for recruiting them are due to the preventive healthcare workers at the Preventive Child Health Care services in The Hague and to the caseworkers at BKK, MEE and Bureau Jeugdzorg in the Hollands Midden region. We thank research assistants Ellen Westhoff, Jasper Boerrigter, Sophie Wins and Rosanne Schoorl for their work on this project, and Yvonne Schonbeck for her support with project organization. Finally, we thank Meinou Theunissen for data analysis and Mascha Kamphuis for management of the project.

## Author Contributions

**Conceptualization:** Noortje M. Pannebakker, Paul L. Kocken, Paula van Dommelen, Krista van Mourik, Ria Reis.

**Funding acquisition:** Paul L. Kocken.

**Investigation:** Noortje M. Pannebakker, Krista van Mourik.

**Methodology:** Noortje M. Pannebakker, Paul L. Kocken, Paula van Dommelen, Sijmen A. Reijneveld, Mattijs E. Numans.

**Project administration:** Noortje M. Pannebakker, Paul L. Kocken, Krista van Mourik.

**Supervision:** Paul L. Kocken.

**Writing – original draft:** Noortje M. Pannebakker, Paul L. Kocken, Sijmen A. Reijneveld, Mattijs E. Numans.

**Writing – review & editing:** Noortje M. Pannebakker, Paul L. Kocken, Ria Reis, Sijmen A. Reijneveld, Mattijs E. Numans.

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
