## [Decision Letter · Decision Letter 0]

19 Nov 2019

PONE-D-19-15380

Care use and its intensity in children with complex problems are related to varying child and family factors: a follow-up study

PLOS ONE

Dear Authors,

Thank you for submitting your manuscript to PLOS ONE. After careful consideration, we feel that it has merit but does not fully meet PLOS ONE’s publication criteria as it currently stands. Therefore, we invite you to submit a revised version of the manuscript that addresses the points raised during the review process.

We would appreciate receiving your revised manuscript by 12/18/2019. To enhance the reproducibility of your results, we recommend that if applicable you deposit your laboratory protocols in protocols.io, where a protocol can be assigned its own identifier (DOI) such that it can be cited independently in the future. For instructions see: http://journals.plos.org/plosone/s/submission-guidelines#loc-laboratory-protocols

We look forward to receiving your revised manuscript.

Kind regards,

Luca Cerniglia, PhD

Academic Editor

PLOS ONE

Journal Requirements:

1. Please include additional information regarding the survey or questionnaire used in the study and ensure that you have provided sufficient details that others could replicate the analyses. For instance, if you developed a questionnaire as part of this study and it is not under a copyright more restrictive than CC-BY, please include a copy, in both the original language and English, as Supporting Information.

Reviewers' comments:

Reviewer's Responses to Questions

**Comments to the Author**

1. Is the manuscript technically sound, and do the data support the conclusions?

Reviewer #1: Partly

Reviewer #2: Partly

2. Has the statistical analysis been performed appropriately and rigorously? 

Reviewer #1: Yes

Reviewer #2: Yes

3. Have the authors made all data underlying the findings in their manuscript fully available?

Reviewer #1: No

Reviewer #2: Yes

4. Is the manuscript presented in an intelligible fashion and written in standard English?

Reviewer #1: Yes

Reviewer #2: No

5. Review Comments to the Author

Reviewer #1: Introduction:

How are “children with complex problems” defined? Is this your definition, or one that has been frequently used in the literature? This is an unfamiliar term to me, and without description, I am uncertain about why this population is of interest or how it differs from other “high-risk” subsets of youth.

I appreciate brevity, but previous literature and how it is insufficient and/or different from the current study, and how it drives the need for the current study, needs a bit more explanation. As does the Anderson and Newman behavioral health model – I appreciate the brief example of enabling factors, for example – but perhaps an additional sentence or two to introduce the model given its importance to the study?

The previously published paper’s findings and how they influenced this study need to be explained. And, what does “more variation in complex problems” mean?

Which types of predisposing, enabling, and need factors will be explored and why (on the basis of what literature?)

How are care services defined? This definition needs to be alluded to even in the intro

Method:

How do you define “specialist” services? In general, the paper contains jargon (“multiproblem” families?)

More information needed on how children were identified for the study. For example, “during well visits” doesn’t describe how they were identified as potential participants, or give info as to rates of declined participation etc. Also, what constituted major life events? What did “persistent parenting concerns” consist of and how were those identified?

Who collected the data; how were digital questionnaires offered?

Were OT and speech services included? (please define “services” more exactly)

Please explain reliability/validity information for selected scales

What is the TIC-P?

Was the parenting concerns question answered dichotomously?

The Method does not provide enough information to support replicability (nor contextualization of study results give the sample – for example, how does the sample studied compare to the population of children with CP? To children in the general population?)

Unless this is a journal specific variation, info about participants should go in Method section

Discussion:

A strength of the study is noted to be the “community-based” sample – please be more specific about how sample was obtained and how it differs from other samples utilized in similar research

In what way do you think recall bias influenced the data? For example, it seems that those who perceive their children’s needs as greater might be more likely to overestimate care use & contacts

Why is limiting care consumption a goal? Often, research on engagement in mental health/psychosocial services has the opposite goal (increase engagement to improve treatment benefit).

Summary:

Research on care use and intensity of use is of importance but results seem to highly differ per study given the sample and the type of care use in question. Thus, my rationale for asking for additional contextualization and information about the sample.

Reviewer #2: In the manuscript entitled “Care use and its intensity in children with complex problems are related to varying child and family factors: a follow-up study”, the authors aimed to investigate mainly the impact of child and family factors on the intensity of care use by children with complex problems (CP), with particular attention to psychological care.

The results of this study are interesting, the manuscript has several positive aspects, but I think that it needs some revisions in order to be clearer to the readers.

In a few cases the text is not clear and I would suggest some English editing.

The introduction, the density of results and the discussion are not always aligned.

The following are suggestions to make the manuscript clearer:

- The Introduction suffers from some weaknesses.

In my opinion, the first part of the Introduction should be made clearer. I think that the main focus of the article would benefit from a more logical structure in which to insert a deeper literature review even if in this field the articles are not many.

For example the authors state “The few studies on this topic show that factors predicting the intensity of use differ from those predicting whether care is used at all”. How does this happen?

In addition at the end of the introduction we find a sentence that is not clear: when they write: “in a study group of children with more variation in complex problems” what do they mean by “more variation in complex problems”?

Finally, the motivation for the study could be explained better, providing explanations to why this particular study is needed and what exact problems it will solve.

- In the Measures section (p. 5), the authors use the acronym TIC-P without ever describing what it corresponds to.

- In the description of the sample, they refer to two groups (the community sample and the sample of care use for complex problems); in the description of the results they mention the number of parents on the two groups at follow-up at 1 year, but they don’t mention the number of persons at the beginning of the study.

One more aspect about the sample requires clarification. It seems that in the subsequent analyses of the predisposing, enabling and need factors the sample is considered a single group… Is this so?

- In the Results section (p.6), describing the sample, the authors state: “Similar percentages had

experienced mental health problems and/or some burden of adverse life events in the previous year”. What do they refer to with “similar percentages”? To “About half of the parents”? This needs to be explained.

- In the Discussion, the authors describe the results of the study, but, in my opinion, they should also interpret them more. Authors should provide a possible explanation of the results.

- Finally, two minor errors to amend:

1. The references should be reviewed because the numbers in the text do not correspond to the final list.

2. In the Discussion section, please, remove the reference to the table “see Table 2” (p. 12). It is not necessary.

6. PLOS authors have the option to publish the peer review history of their article (what does this mean?). If published, this will include your full peer review and any attached files.

Reviewer #1: Yes: Erin Hambrick

Reviewer #2: No

---

## [Author Response · Author response to Decision Letter 0]

23 Mar 2020

Review Comments to the Author

Reviewer #1: 

Introduction:

[COMMENT 1.1]

How are “children with complex problems” defined? Is this your definition, or one that has been frequently used in the literature? This is an unfamiliar term to me, and without description, I am uncertain about why this population is of interest or how it differs from other “high-risk” subsets of youth.

[RESPONSE]

We thank the reviewer for the request for more information about the definition of the target group of our study. Children with complex problems are increasingly being studied because of their acute needs and the challenges facing Western countries in organizing their care. 

We have added the following text (Introduction; new text italicized):

“Little research has been conducted on factors affecting the intensity of care use by children with complex problems (CP). The need of these children for health and social services, especially psychosocial care, is typically greater than would be expected based on their chronic physical, developmental, behavioral or emotional condition; this is the case because their problems interact and enhance vulnerabilities [1-3, Mendenhall et al., 2017]. These children are also referred to as members of troubled families or hotspotters [Hayden & Jenkins, 20145; Hayden & Jenkins, 2015; Gawande 2011]. Children with complex problems form the top 5% of children with the most challenging problems, amounting in the Netherlands to 170,000 children. Western countries struggle to organize effective and efficient care pathways for these children [Tausendfreud et al., 2014]. As a result, a major part of the budgets of psychosocial services is spent on children with CP [Goerge et al., 2010; Sacco et al., 2008].”

Added references:

1. Mendenhall E, Kohrt BA, Norris SA, Ndetei D, Prabhakaran D. Non-communicable disease syndemics: poverty, depression, and diabetes among low-income populations. The Lancet 2017 4–10 March 2017; 389(10072):951-963. 

2. Hayden C, Jenkins C. Children taken into care and custody and the 'troubled families' agenda in England. Child Fam Soc Work 2015;20(4):459-469. 

3. Hayden C, Jenkins C. ‘Troubled Families’ Programme in England: ‘wicked problems’ and policy-based evidence. Policy Stud 2014;35(6):631-649. 

4. Tausendfreund T, Knot-Dickscheit J, Schulze GC, Knorth EJ, Grietens H. Families in multi-problem situations: Backgrounds, characteristics, and care services. Child Youth Serv 2016;37(1):4-22. 

5. Gawande A. The Hot Spotters, can we lower medical costs by giving the neediest patients better care? The New Yorker, January 17th 2011: 40-51. 

6. Goerge, RM, Smithgall, C, Seshadri, R, Ballard, P. llinois Families and Their Use of Multiple Service Systems. Chicago: Chapin Hall at the University of Chicago; 2010. 

7. Sacco FC, Twemlow SW, Fonagy P. Secure attachment to family and community: a proposal for cost containment within high user populations of multiple problem families Smith College Studies in Social Work 2008

[COMMENT 1.2a]

I appreciate brevity, but previous literature and how it is insufficient and/or different from the current study, and how it drives the need for the current study, needs a bit more explanation. 

[RESPONSE]

We thank the reviewer for the request for more information about the relevance of our study. We have added the following text (Introduction; new text italicized):

“The determinants of care use as such have been studied in depth, revealing several factors that impact access [Nanninga et al., 2015; Burns et al., 2004; Verhulst et al., 1997; Tick ea, 2008; Tausendfreud et al., 2016; Pannebakker et al., 2018]. Less attention was paid to understanding the intensity of care use, i.e. the number of contacts with care providers. The scarce literature shows that higher intensity of care-service use by children with CP is related to two main groups of factors: child factors (age and impact of psychosocial problems); and parental factors (educational level, healthcare use, social support, and parental psychosocial problems) [4-6]. Research shows that the determinants affecting care use (yes/no) and the intensity with which it is used differ when studied simultaneously [7-10]. This suggests that intensity of care use may be a unique component of the help-seeking behavior of children with complex problems and their parents. A better understanding of the intensity of care use will help us to organize more efficient care paths for these children.”

Added references:

1. Nanninga M, Jansen DEMC, Knorth EJ, Reijneveld SA. Enrolment of children and adolescents in psychosocial care: more likely with low family social support and poor parenting skills. Eur Child Adolesc Psychiatry 2015;24 (4):407-416. 

2. Burns BJ, Phillips SD, Wagner HR, Barth RP, Kolko DJ, Campbell Y, et al. Mental health need and access to mental health services by youths involved with child welfare: A national survey. J Am Acad Child Adolesc Psychiatry 2004;43(8):960-970. 

3. Verhulst FC, Van Der Ende J. Factors associated with child mental health service use in the community. J Am Acad Child Adolesc Psychiatry 1997;36(7):901-909. 

4. Tick NT, Van Der Ende J, Verhulst FC. Ten-year increase in service use in the Dutch population. European Child and Adolescent Psychiatry 2008;17(6):373-380. 

5. Tausendfreund T, Knot-Dickscheit J, Schulze GC, Knorth EJ, Grietens H. Families in multi-problem situations: Backgrounds, characteristics, and care services. Child Youth Serv 2016;37(1):4-22. 

6. Pannebakker NM, Kocken PL, Theunissen MHC, van Mourik K, Crone MR, Numans ME, et al. Services use by children and parents in multiproblem families. Child Youth Serv Rev 2018;84:222-228. 

[COMMENT 1.2b]

As does the Anderson and Newman behavioral health model – I appreciate the brief example of enabling factors, for example – but perhaps an additional sentence or two to introduce the model given its importance to the study?

[RESPONSE]

We thank the reviewer for the request for a more elaborate introduction of the Andersen framework. We have added the following text (Introduction; new text italicized):

“Research on determinants of care use is often guided by Andersen and Newman’s behavioral-health model [21]. This model was developed to explain the use of care by an individual or population, and has shown its value as comprehensive model for this purpose in health care research during the past decades [Phillips et al., 1998; Babitsch et al., 2012). The model describes care use on the basis of three factors: 1. predisposing factors, i.e., a child’s characteristics or abilities to use a specific service (such as age); 2. enabling factors, i.e., means whereby a family accesses care (such as social support); and 3. healthcare needs (such as a child’s psychosocial problems). This broad framework is a good fit with the wide-ranging problems experienced by children with CP and accommodates a systematic approach to assessing determinants of care intensity. Our study is the first to apply this framework to the intensity of care use by children with CP.”

Added references 

1. Phillips KA, Morrison KR, Andersen R, Aday LA. Understanding the context of healthcare utilization: Assessing environmental and provider-related variables in the behavioral model of utilization. Health Serv Res 1998;33(3 I):571-596.

2. Babitsch B, Gohl D, von Lengerke T. Re-revisiting Andersen's Behavioral Model of Health Services Use: a systematic review of studies from 1998-2011. Psychosoc Med. 2012;9.

[COMMENT 1.3]

The previously published paper’s findings and how they influenced this study need to be explained. And, what does “more variation in complex problems” mean?

[RESPONSE]

We thank the reviewer for her request for additional information about our other article. We have changed the following text (Introduction; new text italicized):

“We previously reported that overall care use was associated with social support and psychosocial problems, and that the use of psychosocial care was also associated with a child’s age and parenting concerns, based on a cross-sectional study in families with severe complex problems [16]. In the current study with a follow-up design, we additionally examined changes in intensity of care use in children with and at risk on developing CP, ensuring a wide range of intensity of care use.”

[COMMENT 1.4]

Which types of predisposing, enabling, and need factors will be explored and why (on the basis of what literature?)

[RESPONSE]

We have added the following text to clarify this (Introduction; new text italicized): 

“We selected several predisposing characteristics of the child (such as age and gender, and also including predetermined factors such as parental education level and adverse life events), as well as enabling (social support and parental care use) and need factors (chronical condition, psychosocial problems, satisfaction with the parent-child relationship, and parenting concerns). This selection was based on the literature regarding determinants impacting the intensity of use of psychosocial care by children, as well as on our former study [4-6; Nanninga et al., 2015; Burns ea 2004; Verhulst et al., 1997; Tick et al., 2008; Tausendfreud et al., 2016; Pannebakker et al., 2018].”

We have added the following text (Introduction; new text italicized):

“The determinants of care use as such have been studied in depth, revealing several factors that impact access (refs). Less attention was focused on understanding the intensity of care use, i.e. the number of contacts with care providers. The scarce literature shows that higher intensity of care-service use by children with CP is related to two main groups of factors: child factors (age and impact of psychosocial problems); and parental factors (educational level, healthcare use, social support, and parental psychosocial problems) [17-20]. Research shows that the determinants affecting care use (yes/no) and the intensity with which it is used differ when studied simultaneously [17-19]. This suggests that intensity of care use may be a unique component of the help-seeking behavior of children with complex problems. A better understanding of intensity of care use will help us to organize more efficient care paths for these children.” 

Added references:

1. Nanninga M, Jansen DEMC, Knorth EJ, Reijneveld SA. Enrolment of children and adolescents in psychosocial care: more likely with low family social support and poor parenting skills. Eur Child Adolesc Psychiatry 2015;24 (4):407-416. 

2. Burns BJ, Phillips SD, Wagner HR, Barth RP, Kolko DJ, Campbell Y, et al. Mental health need and access to mental health services by youths involved with child welfare: A national survey. J Am Acad Child Adolesc Psychiatry 2004;43(8):960-970. 

3. Verhulst FC, Van Der Ende J. Factors associated with child mental health service use in the community. J Am Acad Child Adolesc Psychiatry 1997;36(7):901-909. 

4. Tick NT, Van Der Ende J, Verhulst FC. Ten-year increase in service use in the Dutch population. European Child and Adolescent Psychiatry 2008;17(6):373-380. 

5. Tausendfreund T, Knot-Dickscheit J, Schulze GC, Knorth EJ, Grietens H. Families in multi-problem situations: Backgrounds, characteristics, and care services. Child Youth Serv 2016;37(1):4-22. 

6. Pannebakker NM, Kocken PL, Theunissen MHC, van Mourik K, Crone MR, Numans ME, et al. Services use by children and parents in multiproblem families. Child Youth Serv Rev 2018;84:222-228. 

[COMMENT 1.5]

How are care services defined? This definition needs to be alluded to even in the intro.

[RESPONSE]

We have added the following text as definition of services (Introduction; new text italicized):

“The aim of this study is to identify the changes in the predisposing, enabling and need factors that are associated 1. with a higher likelihood of changes in use of care services and 2. with changes in the intensity of use. The care services use studied comprised a broad spectrum of general care services, including health and psychosocial care, and also a subset psychosocial care, including child mental healthcare and child and family services.”

Method:

[COMMENT 1.6]

How do you define “specialist” services? In general, the paper contains jargon (“multiproblem” families?)

[RESPONSE]

We thank the reviewer for noting our use of jargon. We have replaced multiproblem and defined specialist care (Introduction; new text italicized): 

“in families with severe complex problems [16]”

And 

“The second sample was a sample of care-using children with complex problems. It consisted of families enrolled in specialist child and family services, i.e. services accessible only after referral from primary care.” 

[COMMENT 1.7]

More information needed on how children were identified for the study. For example, “during well visits” doesn’t describe how they were identified as potential participants, or give info as to rates of declined participation etc. Also, what constituted major life events? What did “persistent parenting concerns” consist of and how were those identified?

[RESPONSE]

We thank the reviewer for her request for more information about the sample and procedure. We have made the following changes (Method section, sample and procedure; new text italicized):

“We used the following inclusion procedure. First, a nurse, doctor or social worker identified parents based on our inclusion criteria, which were embedded in their routine intake questionnaire. Professional care givers then provided oral and written information about the study to the identified families and asked permission from parents to be called by a research assistant. Thereafter, the research assistant asked for informed consent regarding participation in this study.”

And

“We included parents when they met the following inclusion criteria: 1. they had a child between 18 months and 12 years and 2. they experienced at least one of the following conditions: A. the child’s elevated total score on the parent-reported Strengths and Difficulties Questionnaire (SDQ) [27] or Brief Infant Toddler Social Emotional Assessment (BITSEA) [28]; B. persistent parenting concerns as judged by the preventive health care worker and/or parents; C. one or more major life event(s) during the past year as assessed using the standard screening questionnaire of the well child clinic [Theunissen, 2019] and D. care utilization of the child or parent in the past six months. Almost all respondents had three or more of these conditions (97%).” 

We moved the following text from results Section to the Methods section: 

A total of 512 parents were approached, 354 of whom participated at T1 (response=69.1%). Of these, 272 participated in the follow-up at 1 year (T1-T2 response =76.8%), 239 from the well-child clinic group (T1=309), and 33 from the group of children using care with high intensity (T1=45). Parents who dropped out at T2 had significantly more sons; more of them were of non-western origin and, on the basis of their home neighborhoods, more of them had a lower socio-economic position [30].”

Added References:

Theunissen, M., Pas van der, S. & Harten van, L. Background information: a uniform protocol for discovering a triage of risks on developmental and health problems by child preventive health care. 2019; 060.16673. 

[COMMENT 1.8] 

Who collected the data; how were digital questionnaires offered?

[RESPONSE]

We have added the following information about the data collection (Method section, sample and procedure; new text italicized):

“Data were collected by trained research assistants at two time points, the first in 2013 (T1) and the second 12 months later (T2). Data were collected in a digital questionnaire, although parents could also opt to be interviewed by telephone in the language of their preference. Parents were reminded three times to fill in this questionnaire and received a gift certificate of 20 euros after doing so. Parents were informed that they could withdraw at any moment.”

[COMMENT 1.9]

Were OT and speech services included? (please define “services” more exactly)

[RESPONSE]

Services are defined as any provider or groups of providers. We measured service use with a preset list of often used services. OT and speech services as such were not in this list; however respondents could endlessly add services to the list. 

We have added to the measurements (Method section; new text italicized):

“Respondents also had the opportunity to add services that we had not listed.”

And

“Services are defined as any provider or group of providers.”

[COMMENT 1.10]

Please explain reliability/validity information for selected scales

[RESPONSE]

We have added the requested information about the reliability/validity of the questionnaires or scales used in the measures: (Method section, new text italicized):

“The children’s service use and intensity of this use in the past six months were measured with the Questionnaire Intensive Care for Youth, a questionnaire measuring use of a pre-set list of types of Dutch services [Bouwmans et al., 2013, Jansen et al., 2015]. This list has been adapted to the setting of care for youth from the valid and reliable Questionnaire for Costs Associated with Psychiatric Illnesses and care use (TIC-P) [Bouwmans et al., 2012; Hakkaart et al., 2002]. As allowed for by this standard questionnaire, we have added and omitted specific items of care services depending on their relevance for our target population. Moreover, respondents had the opportunity to add services we had not listed. Services are defined as any care provider or group of care providers.”

And

“We used validated questionnaires where available, and assessed their reliability in the sample under study.”

And 

“Parental mental health status was measured using the validated 12-item version of the General Health Questionnaire (Cronbach’s α = .86).”

And

“To measure social support, we used two subscales of the validated Dutch Family questionnaire”

And

 “Does your child have one or more chronic health condition—such as asthma, diabetes, ADHD or autism—for which treatment is or was needed? What is the impact of this condition on your child’s daily life?” [Wingerd et al., 2020].”

And

“Finally, parents’ assessment of their relationship with their child was measured using the subscale of the validated Dutch Parenting Load Questionnaire (Cronbach’s α= .83) [44].”

Added references:

1. Bouwmans, CAM, Schawo, SJ, Jansen, DEMC, Vermeulen, KM, Reijneveld, SA, Hakkaart-van Roijen, L. Handleiding Vragenlijst Intensieve Jeugdzorg: Zorggebruik en productieverlies. [Manual Questionnaire Intensive Care for Youth: health care utilization and productivity loss]. Rotterdam: Erasmus MC; 2012. 

2. Jansen DE, Vermeulen KM, Schuurman-Luinge AH, Knorth EJ, Buskens E, Reijneveld SA. Cost-effectiveness of Multisystemic Therapy for adolescents with antisocial behavior: Study protocol of a randomized controlled trial. BMC Public Health 2013;13(1). 

3. Hakkaart-Van Roijen, L, Van Straten, A, Donker, M, Tiemens, B. Trimbos/iMTA questionnaire for Costs associated with Psychiatric Illness (TiC-P). Rotterdam; 2002. 

4. Bouwmans C, De Jong K, Timman R, Zijlstra-Vlasveld M, Van Der Feltz-Cornelis C, Tan SS, et al. Feasibility, reliability and validity of a questionnaire on healthcare consumption and productivity loss in patients with a psychiatric disorder (TiC-P). BMC Health Serv Res 2013;13(1).

5. Wingerd M. Gezondheidsenquete-2013 [Dutch Health care questionnaire]. 12-02-2020; Available at: www.cbs.nl/nl-nl/achtergrond/2015/15/gezondheidsvragenlijst-gezondheidsenquete-2015

[COMMENT 1.11]

What is the TIC-P?

[RESPONSE]

We have added the explanation of the abbreviation TIC-P (Methods section, Measures paragraph, new text italicized): 

“This list has been adapted to the setting of care for youth from the valid and reliable Questionnaire for Costs Associated with Psychiatric Illnesses and care use (TIC-P) [33, 34].”

[COMMENT 1.12]

Was the parenting concerns question answered dichotomously?

[RESPONSE]

Thank you, we have clarified this. We have added the following text (Methods, new text italicized)

“Finally, parents’ assessment of their relationship with their child was measured using the subscale of the Parenting Load Questionnaire (Cronbach’s α= .83) [24]. Answers were given using a five-point Likert-scale. We dichotomized scale sum scores using their medians as cut off.”

[COMMENT 1.13]

The Method does not provide enough information to support replicability (nor contextualization of study results give the sample – for example, how does the sample studied compare to the population of children with CP? To children in the general population?)

[RESPONSE]

We thank the reviewer for her request for additional information about the sample. To our knowledge, no other available study used a Dutch population similar to ours so that we could compare our community based sample. We have added the following text (Methods section, sample and procedure; new text italicized):

“We aimed to include parents of children with or at risk of developing CP, with a wide range in intensity of care use, and living in the community. We recruited these parents from the general population, using inclusion criteria that concurred with the framework for identifying families with complex problems [Bodden & Dekovic, 2010; Bodden& Dekovic, 2015].”

And

“We identified the respondents during well-child visits, which are provided in the Netherlands by the preventive youth healthcare services. Attendance rates at these visits are high: 95% of all children [13]. To ensure the inclusion of children who used care with a high intensity, we additionally included children enrolled in specialist child and family services. Together this study group is expected to represent the whole group of children with CP or at risk of developing CP.”

We also changed (Discussion Section, strengths and limitations; new text italicized)

“Another strength of the study is that the study group of children with or at risk of developing CP were living in the community, including children in treatment with different intensities of care use or not using care at all. In most other research the study groups are limited to children with CP who are using a specific treatment [Bodden & Dekovic, 2015; Tausendfreud et al., 2014].”

Added references:

1. Bodden DHM, Dekovic M. Multiproblem Families Referred to Youth Mental Health: What's in a Name? Fam Process 2016;55(1):31-47. 

2. Tausendfreund T, Knot-Dickscheit J, Schulze GC, Knorth EJ, Grietens H. Families in multi-problem situations: Backgrounds, characteristics, and care services. Child Youth Serv 2016;37(1):4-22. 

[COMMENT 1.14] 

Unless this is a journal specific variation, info about participants should go in Method section

[RESPONSE]

We moved the following text from the Results section to the Methods section: 

“A total of 512 parents were approached, 354 of whom participated at T1 (response=69.1%). Of these, 272 participated in the follow-up at 1 year (T1-T2 response =76.8%), 239 in the community sample, and 33 in the sample of care use for complex problems. Parents who dropped out at T2 had significantly more sons; more of them were also of non-western origin and, on the basis of their home neighborhoods, more of them had a lower socio-economic position [30].”

Discussion:

[COMMENT 1.15]

A strength of the study is noted to be the “community-based” sample – please be more specific about how sample was obtained and how it differs from other samples utilized in similar research.

[RESPONSE]

Thank you; we have clarified this in our response to your COMMENT 1.13, second part. 

[COMMENT 1.16]

In what way do you think recall bias influenced the data? For example, it seems that those who perceive their children’s needs as greater might be more likely to overestimate care use & contacts

[RESPONSE]

We thank the reviewer for challenging us about recall bias. We think intensity of care and its determinants: burden of adverse life events and impact of the chronical condition of the child, are sensitive to recall bias. We think this could lead to either overestimation or underestimation of care use intensity. To our knowledge, no research has been done on the bias in reporting intensity of care use.

We have added to our strengths and limitations (Discussion Section; new text italicized):

“Another limitation is that we used a self-report questionnaire to establish care use in the previous six months. This may have caused some recall bias, especially for intensity of care use and the determinants burden of ALE and impact of chronical conditions of the child. This may have added measurement error and thus a weakening of reported associations, probably without clear under- or overestimation.”

[COMMENT 1.17]

Why is limiting care consumption a goal? Often, research on engagement in mental health/psychosocial services has the opposite goal (increase engagement to improve treatment benefit).

[RESPONSE]

We thank the reviewer for this comment. Goals of our study regard improvement of access and cost-containment. We have clarified this by adding a rationale to the Introduction (Introduction; new text italicized):

“Western countries struggle to organize effective and efficient care pathways for children with CP [Tausendfreud et al., 2014]. As a result, a major part of the budgets of psychosocial services are spent on these children (Goerge et al. 2010; Sacco et al., 2008).”

We have further changed the following text (Discussion section, implications for further research; new text italicized):

“Understanding the mechanism underlying the intensity of care use can help the development of more effective and efficient pathways to care for children with or at risk of developing CP.”

Added references:

1. Tausendfreund T, Knot-Dickscheit J, Schulze GC, Knorth EJ, Grietens H. Families in multi-problem situations: Backgrounds, characteristics, and care services. Child Youth Serv 2016;37(1):4-22. 

2. Goerge, RM, Smithgall, C, Seshadri, R, Ballard, P. llinois Families and Their Use of Multiple Service Systems. Chicago: Chapin Hall at the University of Chicago; 2010. 

3. Sacco FC, Twemlow SW, Fonagy P. Secure attachment to family and community: a proposal for cost containment within high user populations of multiple problem families Smith College Studies in Social Work 2008. 

Summary:

[COMMENT 1.18]

Research on care use and intensity of use is of importance but results seem to highly differ per study given the sample and the type of care use in question. Thus, my rationale for asking for additional contextualization and information about the sample.

[RESPONSE]

We thank the reviewer for her constructive comments. We hope that our responses have adequately provided the requested information and clarifications.

Reviewer #2 (anonymous): In the manuscript entitled “Care use and its intensity in children with complex problems are related to varying child and family factors: a follow-up study”, the authors aimed to investigate mainly the impact of child and family factors on the intensity of care use by children with complex problems (CP), with particular attention to psychological care.

The results of this study are interesting, the manuscript has several positive aspects, but I think that it needs some revisions in order to be clearer to the readers. In a few cases the text is not clear and I would suggest some English editing. The introduction, the density of results and the discussion are not always aligned. 

The following are suggestions to make the manuscript clearer:

[COMMENT 2.0]

I would suggest some English editing

[Response]

The manuscript had already been assessed by a native speaker. A native speaker again checked the full text.

[COMMENT 2.1]

The Introduction suffers from some weaknesses.

In my opinion, the first part of the Introduction should be made clearer. I think that the main focus of the article would benefit from a more logical structure in which to insert a deeper literature review even if in this field the articles are not many.

For example the authors state “The few studies on this topic show that factors predicting the intensity of use differ from those predicting whether care is used at all”. How does this happen?

[RESPONSE]

We thank the reviewer for requesting clarification of the structure of the introduction and more information about the literature review underlying our study. We have added the following text (Introduction; new text italicized):

“The determinants of care use as such have been studied in depth, revealing several factors that impact access [Nanninga et al., 2015; Burns et al., 2004; Verhulst et al., 1997; Tick ea, 2008; Tausendfreud et al., 2016; Pannebakker et al., 2018]. Less attention was paid to understanding the intensity of care use, i.e. the number of contacts with care providers. The scarce literature shows that higher intensity of care-service use by children with CP is related to two main groups of factors: child factors (age and impact of psychosocial problems); and parental factors (educational level, healthcare use, social support, and parental psychosocial problems) [4-6]. Research shows that the determinants affecting care use (yes/no) and the intensity with which it is used differ when studied simultaneously [7-10]. This suggests that intensity of care use may be a unique component of the help-seeking behavior of children with complex problems and their parents. A better understanding of the intensity of care use will help us to organize more efficient care paths for these children.”

Added references:

1. Nanninga M, Jansen DEMC, Knorth EJ, Reijneveld SA. Enrolment of children and adolescents in psychosocial care: more likely with low family social support and poor parenting skills. Eur Child Adolesc Psychiatry 2015;24 (4):407-416.

2. Burns BJ, Phillips SD, Wagner HR, Barth RP, Kolko DJ, Campbell Y, et al. Mental health need and access to mental health services by youths involved with child welfare: A national survey. J Am Acad Child Adolesc Psychiatry 2004;43(8):960-970. 

3. Verhulst FC, Van Der Ende J. Factors associated with child mental health service use in the community. J Am Acad Child Adolesc Psychiatry 1997;36(7):901-909. 

4. Tick NT, Van Der Ende J, Verhulst FC. Ten-year increase in service use in the Dutch population. European Child and Adolescent Psychiatry 2008;17(6):373-380. 

5. Tausendfreund T, Knot-Dickscheit J, Schulze GC, Knorth EJ, Grietens H. Families in multi-problem situations: Backgrounds, characteristics, and care services. Child Youth Serv 2016;37(1):4-22. 

6. Pannebakker NM, Kocken PL, Theunissen MHC, van Mourik K, Crone MR, Numans ME, et al. Services use by children and parents in multiproblem families. Child Youth Serv Rev 2018;84:222-228. 

[COMMENT 2.2]

In addition at the end of the introduction we find a sentence that is not clear: when they write: “in a study group of children with more variation in complex problems” what do they mean by “more variation in complex problems”?

[RESPONSE]

We thank the reviewer for her request for additional information about this previous study. We have changed the following text (Introduction; new text italicized):

“We previously reported that overall care use was associated with social support and psychosocial problems, and that the use of psychosocial care was also associated with a child’s age and parenting concerns, based on a cross-sectional study in families with severe complex problems [16]. In the current study, using a follow-up design, we examined changes in intensity of care use, as well as changes in the use of care itself, both for children with and at risk of developing CP, to ensure covering a wide range of intensity of care use.”

[COMMENT 2.3]

Finally, the motivation for the study could be explained better, providing explanations to why this particular study is needed and what exact problems it will solve.

[RESPONSE]

We thank the reviewer for requesting more information about the knowledge gap we hope to fill with our study. We have added the following text (Introduction; new text italicized):

“Little research has been conducted on factors affecting the intensity of care use by children with complex problems (CP). The need of these children for health and social services, especially psychosocial care, is typically greater than would be expected based on their chronic physical, developmental, behavioral or emotional condition; this is the case because their problems interact and enhance vulnerabilities [1-3, Mendenhall et al., 2017]. These children are also referred to as members of troubled families or hotspotters [Hayden & Jenkins, 20145; Hayden & Jenkins, 2015; Gawande 2011; Stanhope et al., 2015]. Children with complex problems form the top 5% of children with the most challenging problems, amounting in the Netherlands to 170,000 children. Western countries struggle to organize effective and efficient care pathways for these children [Tausendfreud et al., 2014]. As a result, a major part of the budgets of psychosocial services is spent on children with CP [Goerge et al., 2010; Sacco et al., 2008].”

Added references

1. Mendenhall E, Kohrt BA, Norris SA, Ndetei D, Prabhakaran D. Non-communicable disease syndemics: poverty, depression, and diabetes among low-income populations. The Lancet 2017 4–10 March 2017; 389(10072):951-963. 

2. Hayden C, Jenkins C. Children taken into care and custody and the 'troubled families' agenda in England. Child Fam Soc Work 2015;20(4):459-469. 

3. Hayden C, Jenkins C. ‘Troubled Families’ Programme in England: ‘wicked problems’ and policy-based evidence. Policy Stud 2014;35(6):631-649. 

4. Tausendfreund T, Knot-Dickscheit J, Schulze GC, Knorth EJ, Grietens H. Families in multi-problem situations: Backgrounds, characteristics, and care services. Child Youth Serv 2016;37(1):4-22. 

5. Gawande A. The Hot Spotters, can we lower medical costs by giving the neediest patients better care? The New Yorker, January 17th 2011: 40-51. 

Measures section 

[COMMENT 2.4]

- In the (p. 5), the authors use the acronym TIC-P without ever describing what it corresponds to.

[RESPONSE]

We have explained the abbreviation TIC-P (Methods section, Measures paragraph, new text italicized): 

“Questionnaire for Costs Associated with Psychiatric Illnesses and Care Use (TiC-P) [33, 34].”

[COMMENT 2.5]

- In the description of the sample, they refer to two groups (the community sample and the sample of care use for complex problems); in the description of the results they mention the number of parents on the two groups at follow-up at 1 year, but they don’t mention the number of persons at the beginning of the study.

[RESPONSE]

We have added the number of parents at the beginning of the study per subgroup (Methods, new text italicized): 

“A total of 512 parents were approached, 354 of whom participated at T1 (response=69.1%). Of these, 272 participated in the follow-up at 1 year (T1-T2 response =76.8%), 239 from the well-child clinic group (T1=309) and 33 from the group of children using care with a high intensity (T1=45).”

[COMMENT 2.6]

One more aspect about the sample requires clarification. It seems that in the subsequent analyses of the predisposing, enabling and need factors the sample is considered a single group… Is this so?

[RESPONSE]

Yes, the sample is considered as one group of children coming from the same community. We hope to have clarified this by our modifications in the Methods section (Methods section, sample and procedure; new text italicized):

“We aimed to include parents of children with or at risk of developing CP, with a wide range in intensity of care use, and living in the community. We recruited these parents from the general population, using inclusion criteria that concurred with the framework for identifying families with complex problems [Bodden & Dekovic, 2010; Bodden & Dekovic, 2015]. We identified the respondents during well-child visits, which are provided in the Netherlands by the preventive youth healthcare services. Attendance rates at these visits are high: 95% of all children [29]. To ensure the inclusion of children who used care with a high intensity, we additionally included children enrolled in specialist child and family services. Together this study group is expected to represent the whole group of children with CP or at risk of developing CP.”

We also changed (Discussion Section, strengths and limitations; new text italicized)

“Another strength of the study is that the study group of children with or at risk of developing CP were living in the community, including children in treatment with different intensity of care use or not using care at all. In most other research the study groups are limited to children with CP who are using a specific treatment [Bodden & Dekovitch, 2015; Tausendfreud et al., 2014].“

Added references:

1. Bodden DHM, Dekovic M. Multiproblem Families Referred to Youth Mental Health: What's in a Name? Fam Process 2016;55(1):31-47. 

2. Tausendfreund T, Knot-Dickscheit J, Schulze GC, Knorth EJ, Grietens H. Families in multi-problem situations: Backgrounds, characteristics, and care services. Child Youth Serv 2016;37(1):4-22. 

RESULTS

[COMMENT 2.7]

In the Results section (p.6), describing the sample, the authors state: “Similar percentages had

experienced mental health problems and/or some burden of adverse life events in the previous year”. What do they refer to with “similar percentages”? To “About half of the parents”? This needs to be explained.

[RESPONSE]

“Similar percentages” indeed refers to the earlier mentioned “about half of the parents”. 

We have changed this to (Results section, background characteristics; new text italicized):

“About half of the parents with a high educational level experienced mental health problems and/or experienced burden of adverse life events in the previous year.” 

DISCUSSION

[COMMENT 2.8]

 In the Discussion, the authors describe the results of the study, but, in my opinion, they should also interpret them more. Authors should provide a possible explanation of the results.

[RESPONSE]

We thank the reviewer for challenging us to more fully interpret and explain our results. We have tried to be more specific in our interpretations. We have also divided the second paragraph of the Discussion into two separate ones. The modified text is (Discussion; new text italicized): 

“We found that several changes in predisposing (i.e. burden of ALE and a child’s age) and need factors (i.e. parenting concerns and psychosocial problems) were associated with changes in care use and its intensity, both for overall care use and for the use of psychosocial services, but that changes in enabling factors were not. The determinants found are in line with previous findings [10, 12, 33-36]. An explanation may be that enabling factors are harder to change than predisposing and need factors in the relative short time span of our study (one year). For example, it is more difficult for a child social worker to convince parents to make use of mental health care for their own mental problems than to address parenting concerns. This study shows the value of the Andersen and Newman model in studying the intensity of care use, especially in distinguishing enabling factors from other factors that affect families with a child at risk of CP or at risk of developing CP. 

The results of this study added the burden of ALE as a factor impacting change in intensity of care use. Research has showed that ALE is an important determinant of care use in general [56,67]. ALE will especially affect children with CP, interacting strongly with the other problems of these children, thereby leading to more intense problems. Unexpectedly, we found a slight negative relative risk between ALE and the intensity of any care use. We noted that the burden of ALE decreased in a relatively large group of children while they were using care. Children with CP may have been motivated to continue treatment even when the burden of ALE decreased, because trauma-based therapies are known to have a positive effect on other emotional conditions [Schneider et al., 2012; Connor et al., 2015]. Furthermore, when the safety of a child is at risk, as in cases of domestic violence, care professionals will ideally continue treatment to monitor the situation. Our results indicate that changes in ALE are relevant to the whole care process, i.e. not only to care use itself, but also its intensity.

 Although improving social support is at the core of treatment of families with complex problems, in the final models of our study this factor was absent [Stanhope et al.,2015]. It can be hypothesized that social support works differently for families with complex problems than for the general population [Matos et al., 2004; Sousa et al., 2005; Sousa et al., 2009, Pannebakker et al., 2018]. Usually in cases of children with CP, the families’ social networks are large and suitable for dealing with the challenges of daily living [Matos et al., 2004; Sousa et al., 2005; Sousa et al., 2009]. However, regardless of their perceived social support, families will turn to professionals to bring about long-term improvements, surmising that they may not be able to achieve these improvements with their own network. Also, professionals may not yet have managed to bring about changes in the quality of support by the social environment because of the relatively short period of our study (one year). Although social support is a known determinant of a child’s care use, more research is needed to understand how to optimize its impact for families with a child with CP.”

Finally, we have added the following text to the implications (Discussion section, implications for practice; new text italicized):

“Regarding the enabling factor social support, more research is needed on how to improve the quality of support provided by the network of families with a child with complex problems. We advise the development of a treatment module to improve support skills for parents and the key persons of their social network. These new skills can be taught by volunteers who are able to model healthy support.”

Added references

1. Schneider SJ, Grilli SF, Schneider JR. Evidence-based treatments for traumatized children and adolescents. Curr Psychiatry Rep 2013;15(1).-012-0332-5.

2. Connor DF, Ford JD, Arnsten AFT, Greene CA. An update on posttraumatic stress disorder in children and adolescents. Clin Pediatr 2015;54(6):517-528. 

3. Stanhope V, Videka L, Thorning H, McKay M. Moving Toward Integrated Health: An Opportunity for Social Work. Soc Work Health Care 2015;54(5):383-407. 

4. Matos AR, Sousa LM. How multiproblem families try to find support in social services. J Soc Work Pract 2004;18(1):65-80. 

5. Sousa L. Building on personal networks when intervening with multi-problem poor families. J Soc Work Pract 2005;19(2):163-179. 

6. Sousa L, Rodrigues S. Linking formal and informal support in multiproblem low-income families: The role of the family manager. J Community Psychol 2009;37(5):649-662. 

7. Pannebakker NM, Kocken PL, Theunissen MHC, van Mourik K, Crone MR, Numans ME, et al. Services use by children and parents in multiproblem families. Child Youth Serv Rev 2018;84:222-228. 

[COMMENT 2.9]

Finally, two minor errors to amend:

1. The references should be reviewed because the numbers in the text do not correspond to the final list.

2. In the Discussion section, please, remove the reference to the table “see Table 2” (p. 12). It is not necessary.

[RESPONSE]

Thank you for noticing these minor errors. We reviewed the references and removed the reference “see Table 2” from the Discussion section.

---

## [Editor Report · Decision Letter 1]

30 Mar 2020

Care use and its intensity in children with complex problems are related to varying child and family factors: a follow-up study

PONE-D-19-15380R1

Dear Authors,

We are pleased to inform you that your manuscript has been judged scientifically suitable for publication and will be formally accepted for publication once it complies with all outstanding technical requirements.

With kind regards,

Luca Cerniglia, PhD

Academic Editor

PLOS ONE

Additional Editor Comments (optional):

The Authors were responsive to all comemnts and I think the paper can be accepted in the present version.
---

## [Editor Report · Acceptance letter]

9 Apr 2020

PONE-D-19-15380R1 

Care use and its intensity in children with complex problems are related to varying child and family factors: a follow-up study 

Dear Dr. Pannebakker:

I am pleased to inform you that your manuscript has been deemed suitable for publication in PLOS ONE. Congratulations! Your manuscript is now with our production department. 

With kind regards,

on behalf of

Dr. Luca Cerniglia 

Academic Editor

PLOS ONE